# General synthesis of ionic-electronic coupled two-dimensional materials

Xiang Xu[1,4], Yunxin Chen[1,4], Pengbin Liu[1], Hao Luo[2], Zexin Li[1], Dongyan Li[1], Haoyun Wang[1], Xingyu Song[1], Jinsong Wu [2], Xing Zhou [1] ✉ & Tianyou Zhai [1,3] ✉

Two-dimensional (2D) $AMX_2$ compounds are a family of mixed ionic and electronic conductors (where A is a monovalent metal ion, M is a trivalent metal, and X is a chalcogen) that offer a fascinating platform to explore intrinsic coupled ionic-electronic properties. However, the synthesis of 2D $AMX_2$ compounds remains challenging due to their multielement characteristics and various by-products. Here, we report a separated-precursor-supply chemical vapor deposition strategy to manipulate the chemical reactions and evaporation of precursors, facilitating the successful fabrication of 20 types of 2D $AMX_2$ flakes. Notably, a 10.4 nm-thick $AgCrS_2$ flake shows superionic behavior at room temperature, with an ionic conductivity of 192.8 mS/cm. Room temperature ferroelectricity and reconfigurable positive/negative photovoltaic currents have been observed in $CuScS_2$ flakes. This study not only provides an effective approach for the synthesis of multielement 2D materials with unique properties, but also lays the foundation for the exploration of 2D $AMX_2$ compounds in electronic, optoelectronic, and neuromorphic devices.

The coupled ionic-electronic effects in two-dimensional (2D) materials have attracted tremendous interest in recent years as they endow the materials with diverse responses to external stimuli, further facilitating the development of next-generation electronic, optoelectronic, and neuromorphic devices[1–5]. For example, the hybrid of the ionic gate with 2D materials enables the modulation of the phase transition[6–8] and band structures[9–11] in 2D materials due to the strong gate control ability of the ionic gate. Furthermore, the extrinsic ionic states can be introduced into 2D materials through the pre-treatment, such as the intercalation of external ions[4,12] and plasma treatment[5,13]. Subsequently, employing an electric field to control the migration of ions allows for emulating the function of biological neurons and synapses, showing the vast potential in the field of neuromorphic computing[4,5,13]. It should be noted that these additional modification techniques require complex processes and result in interface states[4,11], thereby impeding the exploration of novel physical and chemical properties, as

well as hindering the development of high-density integrated devices. The intrinsic ionic-electronic coupled 2D materials are expected to solve the above problems.

$AMX_2$ is a family of mixed ionic-electronic conductors (where A is a monovalent metal ion, M is a trivalent transition or main group metal, and X is a chalcogen). The monovalent metal ions $Cu^+$ and $Ag^+$ have 3d-orbital electrons that exhibit second-order Jahn-Teller effect[14], and normally possess a low ion migration barrier[15]. Thus, introducing the superionic conductivity[16–18], multiferroics[19,20], and magnetism[21] properties within the $AMX_2$. Meanwhile, the multielement characteristic and various atomic structures give the $AMX_2$ rich band structures[19,22,23], making the $AMX_2$ compounds excellent systems for studying intrinsic coupled ionic-electronic properties. While a few demonstrations of the synthesis of 2D $AMX_2$ have been reported[16,24], the fabrication of most of these compounds remains elusive, hindering their exploration and application. Chemical vapor deposition (CVD) has been widely used in

[1]State Key Laboratory of Materials Processing and Die & Mould Technology, School of Materials Science and Engineering, Huazhong University of Science and Technology, Wuhan 430074, P. R. China. [2]Nanostructure Research Center, State Key Laboratory of Advanced Technology for Materials Synthesis and Processing, Wuhan University of Technology, Wuhan 430070, P. R. China. [3]Optics Valley Laboratory, Hubei 430074, P. R. China. [4]These authors contributed equally: Xiang Xu, Yunxin Chen. ✉e-mail: zhoux0903@hust.edu.cn; zhaity@hust.edu.cn

the synthesis of 2D materials[25–34]. For the synthesis of multielement compounds like $AMX_2$, three kinds of precursors are required. In the common CVD process, the metal and chalcogen precursors transport along the same path, leading to uncontrollable pre-reactions and an unstable supply of precursors, thereby hindering the controllable synthesis of 2D $AMX_2$.

In this work, we demonstrate a separated-precursor-supply strategy in which the suppressed by-reactions and controllable supply of precursors ensure the general synthesis of 20 distinct 2D $AMX_2$, 18 types of which have never been reported. Interestingly, the as-grown $AMX_2$ flakes exhibit unique electronic and ionic properties. A 10.4 nm $AgCrS_2$ flake shows superionic conductor characteristics at room temperature with an ionic conductivity of up to 192.8 mS/cm. The as-grown $CuScS_2$ flakes exhibit semiconductor ferroelectric properties, and show a Curie temperature reaching ~370 K. Notably,

the reconfigurable positive/negative photovoltaic current can be observed in $CuScS_2$ devices due to the adjustable ion migration drived by the external electric field. This work not only provides an effective strategy for synthesizing multielement 2D materials but also opens up opportunities for studying the properties and potential applications of a wide variety of 2D $AMX_2$.

## Results and discussion

### General growth and characterization of $AMX_2$

We first discuss the difficulties of controllable synthesis 2D $AMX_2$. For multielement compounds like $AMX_2$, during the CVD process, there are many possible reactions between metal and chalcogen precursors (Supplementary Fig. 1). And the formation energy of most binary products is less than that of $AMX_2$ (Fig. 1a)[35]. Even if the $AMX_2$ is more favorable thermodynamically, it is hard to control the

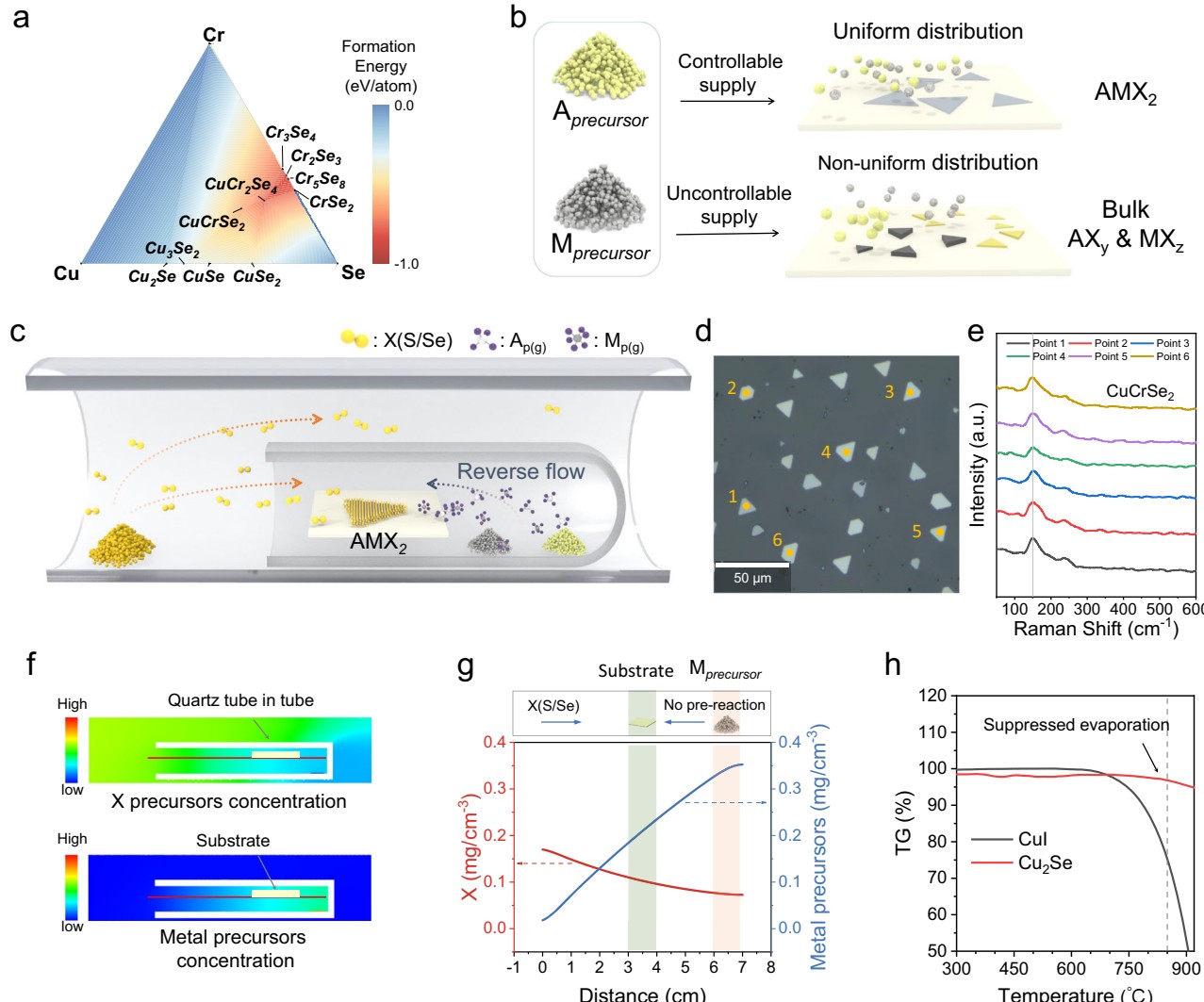

**Fig. 1 | Synthesis mechanism of chemical vapor deposition (CVD) growth $AMX_2$.**
**a** The formation energy of $CuCrSe_2$ phase diagram[35]. **b** The kinetic growth process is influenced by the supply of metal precursors. A and M present the two kinds of metal elements of $AMX_2$ compounds, and X represents the chalcogen element. The x and y in the $AX_y$ and $MX_z$ demonstrate the possible stoichiometric ratio of the binary by-products in the (**a**) (such as CuSe, $Cu_2Se$, etc.). **c** Schematic image of the CVD setup. The orange and blue dash arrows represent the transportation paths of the vapored chalcogen precursor and metal precursors ($A_{p(g)}$ and $M_{p(g)}$), respectively. **d** The large area optical image of the as-synthesized $CuCrSe_2$ nanosheets. **e** The Raman spectra of the flakes in the (**d**). The vertical dash line located at

150 cm$^{-1}$ demonstrates the consistent Raman peaks of the as-synthesized $CuCrSe_2$ nanosheets. **f** The computational fluid dynamics (CFD) simulated distribution of X (S/Se) and metal precursors concentration. **g** The CFD simulated variation curve of precursor concentration along the red line in the (**f**). The green and pink shaded areas schematically represent the position of the substrate and metal precursors, respectively. **h** Thermogravimetric analysis (TGA) of CuI and $Cu_2Se$ powders. The black and red curves correspond to the weight-loss curves of CuI and $Cu_2Se$, respectively. The vertical dash line located at 850°C demonstrates that the evaporation of excessively selenated metal precursors will be significantly suppressed.

reactions of precursors during transportation due to the premixing of the vaporized precursors in common CVD, which promotes the non-uniform distribution of precursors and results in undesired products (Fig. 1b). Taking the synthesis of $CuCrSe_2$ as an example since other $AMX_2$ compounds have similar troubles. In the common CVD process, the Se vapor will pass by the Cu and Cr precursors before reaching the substrate, resulting in uncontrollable pre-reactions (Supplementary Fig. 2). Due to the consistent exposure of the metal source to the Se vapor, the metal precursor powders undergo excessive selenization (Supplementary Fig. 3a), which will suppress the vaporization and destabilize the precursor supply, then giving rise to a large number of by-products such as $Cu_xSe$ on the substrate (Supplementary Fig. 3b, c), hindering the controlled synthesis of $AMX_2$.

To achieve the controllable synthesis of 2D $AMX_2$ compounds, it is imperative to suppress undesired by-reactions. We have approached this challenge from a kinetic perspective. Specifically, we report a separated-precursor-supply strategy to suppress the by-reactions during mass transportation. The schematic representation of the CVD setup can be found in Fig. 1c and Supplementary Fig. 4a. First, stable source feeding of chalcogen is important. Here, we placed the resolidified chalcogen source, which is believed to realize stable source feeding and further reduce chalcogen vacancy forming in the CVD process[36], at the upstream. More importantly, we should ensure the temporal and spatial uniform supply of two metal precursors to support the synthesis of 2D $AMX_2$, which is much more difficult than the synthesis of binary compounds. We placed two kinds of metal precursors at the bottom of a one-side-sealed quartz tube, then the small quartz tube was placed downstream of the furnace tube (see Methods for more details). In this system, the transport process of metal precursors is protected by the small quartz tube and is separated from the transport process of chalcogen vapor (Supplementary Fig. 4b). Based on this method, we realized the uniform synthesis of the 2D $AMX_2$ materials (Fig. 1d, e).

Computational fluid dynamics (CFD) simulations predict that the gas flow inside the small quartz tube is primarily directed towards the open side, opposing the flow direction of carrier gas, and exhibits significantly lower velocity compared to the external gas flow outside the small tube (Supplementary Fig. 6). This results in a reverse mass flow opposite to the Se precursor's transport direction (Supplementary Figs. 6 and 7). As a consequence, the concentration of metal precursor vapors is lower at the tube's open side and higher at the tube's sealed side, and the Se vapors' concentration distribution is opposite to that of the metal precursor vapor (Fig. 1f, g). The relatively high concentration of the metal precursor vapor in the small tube prevents excessive selenization of the metal precursor powder, ensuring the stable vaporization of the metal precursor during the whole CVD process. In contrast, without the confinement of the small quartz tube, the vapor of the metal precursor and Se precursor meet before reaching the substrate (Supplementary Fig. 8) which will facilitate the occurrence of by-reactions. And the concentration of chalcogen precursor is much higher than the metal precursors. This will result in excessive selenization or sulfurization, making the supply of metal precursor unstable (Fig. 1h). However, when the metal precursor is confined within the small quartz tube, its concentration is one order of magnitude higher compared to the situation without spatial confinement (Supplementary Fig. 9). The ample metal precursor supply, which matches the supply of chalcogen precursor, greatly suppress phase separation.

Taking the synthesis of 2D $CuCrSe_2$ as an example again. Different from the common CVD, the separated-precursor-supply strategy protected the metal precursors from excessive selenization and suppressed the pre-reactions between the Cu/Cr precursors and Se precursors (Supplementary Fig. 10), thereby ensuring a stable supply of metal precursors and suppressing undesired by-products. Through this approach, 2D $CuCrSe_2$ with consistent phase and uniform morphology can be obtained (Fig. 1d, e). Figure 2 shows a summary of optical images of the 20 kinds of 2D $AMX_2$ materials prepared using this method. The corresponding synthesis conditions are described in the Methods, more details are summarized in Supplementary Table 1 and Supplementary Methods. To our best knowledge, 18 of them have not been previously synthesized using CVD or mechanical exfoliation methods (Supplementary Table 2). The synthesized $AMX_2$ compounds are mainly selenides and sulfides and contain 9 metal elements including two monovalent metal ions ($Cu^+$, $Ag^+$); three transition metals (Sc, Cr, Fe); and four main group metals (Ga, In, Sb, Bi). To show our uniform growth, the larger area optical images with more flakes are shown in Supplementary Fig. 11. Most of the synthesized 2D $AMX_2$ compounds exhibit triangular or hexagonal shapes, and a small fraction shows rhombic or nanoribbon morphologies. The thickness of most samples can be reduced to below 10 nm, and some can even reach few unit-cell thickness, such as $CuCrS_2$ (2.56 nm), $AgCrSe_2$ (1.86 nm), $CuFeSe_2$ (1.9 nm), and $CuSbS_2$ (0.79 nm) (Supplementary Fig. 12), which demonstrates the effectivity and generality of our growth methods.

To elucidate the structural features of $AMX_2$ compounds, we conducted high-angle annular dark-field scanning transmission electron microscopy (HAADF-STEM) characterization on three representative materials, namely $AgCrS_2$ with $R3m$ space group, $AgBiSe_2$ with $R\bar{3}m$ space group, and $CuInS_2$ with $I\bar{4}2d$ space group. For the as-grown $AgCrS_2$. The High-resolution transmission electron microscopy (HRTEM) image along the [001] crystal direction reveals its hexagonal atomic arrangement, with a measured lattice spacing of $d(2\bar{1}0) = 1.71$ Å (Fig. 3a). The cross-sectional HAADF-STEM image along the [100] crystal direction is shown in Fig. 3b, with a measured lattice spacing of $d(003) = 6.83$ Å. Based on the layered characteristics, the structure can be understood as alternating stacking of $CrS_2$ layers and $Ag^+$ ion layers along the c-axis. The brightness variation of the dashed line in Fig. 3b is depicted in Fig. 3c, where Ag exhibits the highest brightness, followed by Cr with intermediate brightness, and S appears the darkest. The 3d orbital electrons of $Cr^{3+}$ in the material hybridize with the p orbital electrons of S, forming $[CrS_6]$ octahedral coordination structure (shown at the blue quadrilateral position in Fig. 3b). The $[CrS_6]$ octahedra are edge-connected to form the $CrS_2$ layer (Supplementary Fig. 13). Meanwhile, $Ag^+$ ions orderly occupy the tetrahedral sites between the $CrS_2$ layers (shown at the red triangular position in Fig. 3b). This ordered tetrahedral occupancy results in the breaking of inversion symmetry, leading the material to exhibit a pronounced optical second harmonic generation (SHG) response (Supplementary Fig. 14g). Similarly, $AgBiSe_2$ also exhibits typical layered structure characteristics, with $Ag^+$ confined between the $BiSe_2$ layers. The HRTEM image along the [001] crystal direction also demonstrates the characteristic hexagonal atomic arrangement, with a measured lattice spacing of $d(2\bar{1}0) = 2.07$ Å (Fig. 3d). However, different with the $AgCrS_2$, the $Ag^+$ ions in the $AgBiSe_2$ occupy the octahedral sites between the $BiSe_2$ layers (shown at the red quadrilateral in Fig. 3e), with a measured lattice spacing of $d(003) = 6.77$ Å. The structure of $CuInS_2$ is composed of $[CuS_4]$ and $[InS_4]$ tetrahedra. The exposed surface of the sample has a hexagonal atomic arrangement, namely the (112) plane (Fig. 3g). From HRTEM images and selected area electron diffraction (SAED) patterns, the annotated lattice spacings are $d(20\bar{4}) = 1.96$ Å and $d(112) = 3.26$ Å, respectively. The measured crystal plane spacings for these three materials are consistent with previous reports. Additionally, we provided Raman spectra, photoluminescence spectra, and optical SHG response for each sample (Supplementary Figs. 14–18), and we conducted HRTEM and energy-dispersive spectroscopy (EDS) to characterize the 20 kinds of as-grown $AMX_2$ compounds (Supplementary Figs. 19–38). The synthesized $AMX_2$ compounds exhibit good agreement with the expected phases and show high crystalline quality.

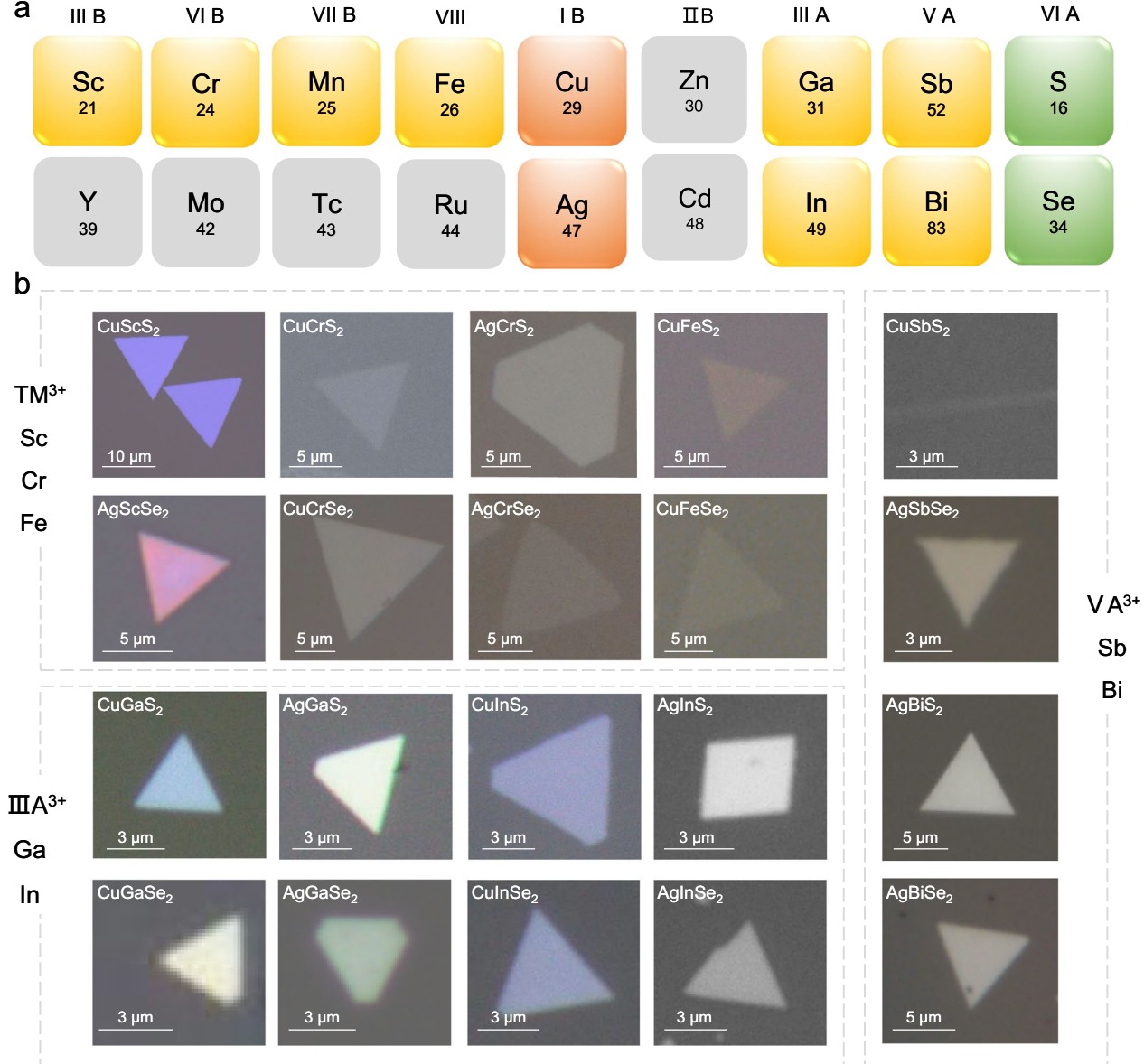

**Fig. 2 | Optical microscopy images of the 20 kinds of as-synthesized 2D AMX$_2$ compounds. a** Summary of 2D AMX$_2$ compounds that can be synthesized using this method. Orange, element A; yellow, element M; green, element X. **b** Optical images of the as-synthesized 2D AMX$_2$ nanosheets.

In summary, the as-grown AMX$_2$ compounds typically possess a quasi-2D layered structure, where monovalent metal ions Cu$^+$ and Ag$^+$, known for their strong migration characteristics[17,18], are confined within the interlayer space of MX$_2$ (Supplementary Fig. 13). This means that most AMX$_2$ materials possess inherent migratable ions, and have natural 2D ion migration pathways. As a result, AMX$_2$ materials exhibit intrinsic ion characteristics[18,23,37]. Additionally, the 3d orbital electrons of the transition metal in the MX$_2$ layers may introduce ferromagnetic or antiferromagnetic characteristics to the materials[19,20,38] (Supplementary Table 3). In addition to their layered structure characteristics, 14 kinds of the as-grown 2D AMX$_2$ possess features with broken inversion symmetry (Supplementary Table 4), endowing them with optical SHG properties, as well as piezoelectric and ferroelectric properties[24,39].

## The ionic and electronic properties of AMX$_2$ compounds

Building upon the structural attributes of AMX$_2$, Cu$^+$/Ag$^+$ ions can undergo hopping between the tetrahedral or octahedral sites within the MX$_2$ interlayer space when the temperature is higher than a certain point[18,37]. Simultaneously, under the influence of an external electric field, ions can exhibit directed migration, thereby manifesting super-ionic conductor features[16,40] (Fig. 4a). We commence our investigation by delving into the ionic migration properties of AMX$_2$, using AgCrS$_2$ as an illustrative example, we fabricated two-terminal electrode devices and employed Au as a blocking electrode for testing ion conductivity. The impedance spectra of samples with different thicknesses (Fig. 4b) can all be fitted with two semicircles. These curves exhibit characteristic mixed ion-electron conductivity features[16,23]. According to the equivalent circuit of the mixed ionic-electronic conductor model (illustrated in the inset of Fig. 4b), ionic conductivity can be obtained by fitting the electrochemical impedance curves, and the detail of the fitting process is described in Methods and Supplementary Fig. 39. In a 10.4 nm AgCrS$_2$ nanosheet, we measured an ionic conductivity of 192.8 mS/cm. Interestingly, we observed that the ionic conductivity increases as the sample thickness decreases (Supplementary Fig. 40). This trend is consistent with the previous report on AgCrS$_2$ samples obtained via electrochemical exfoliation[16]. However, our samples exhibit a higher ionic conductivity, which could be attributed to the superior crystalline quality resulting from our

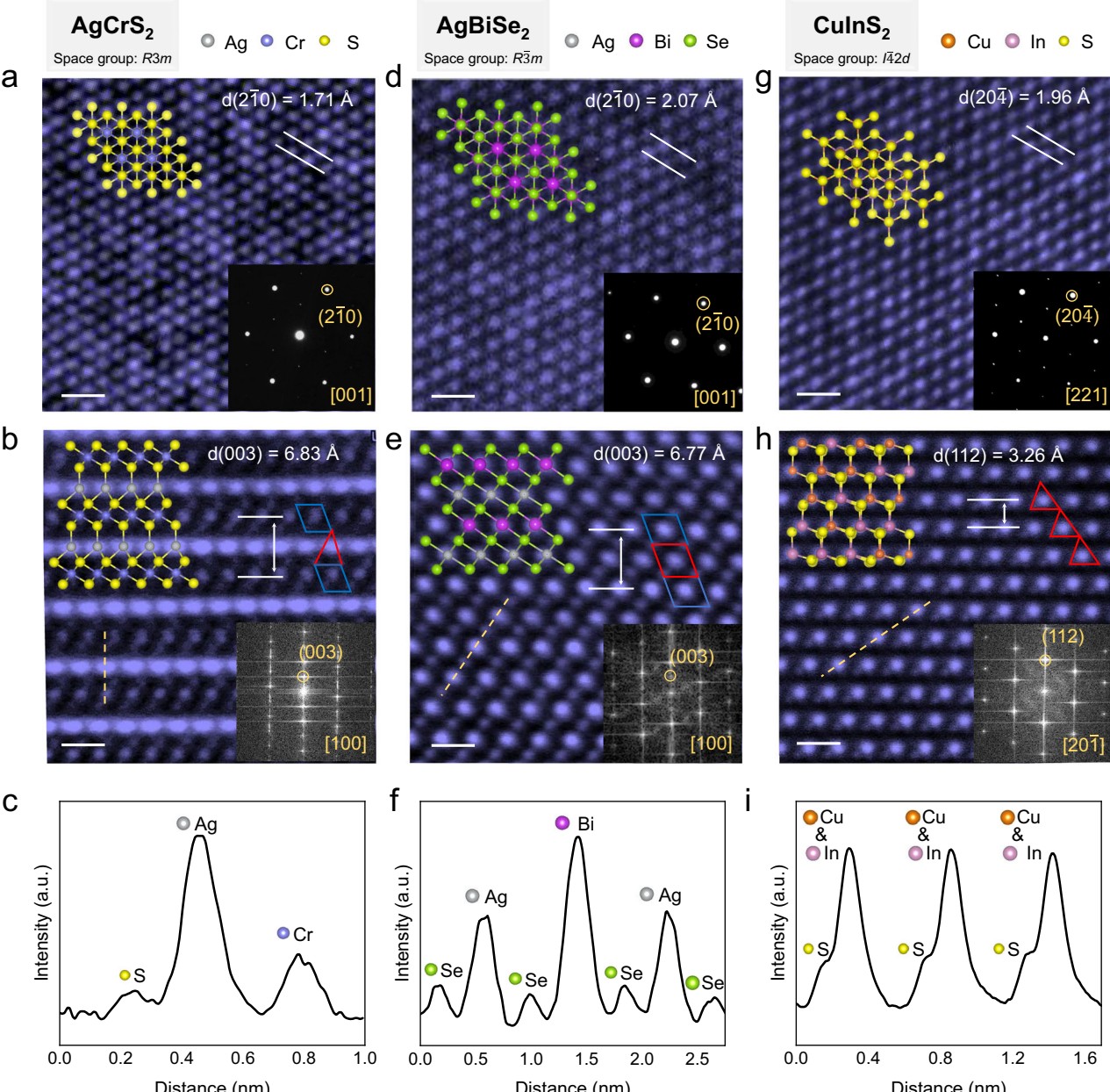

**Fig. 3 | Structural analysis of selected three as-synthesized 2D AMX₂ compounds. a, d, g** High-resolution transmission electron microscopy (HRTEM) images of AgCrS₂, AgBiSe₂, and CuInS₂ along the direction out of the plane of the as-grown nanosheets, scale bar: 0.5 nm. The insets display the top view of atomic structure models and selected area electron diffraction (SAED) patterns. **b, e, h** Cross-sectional high-angle annular dark-field scanning transmission electron microscopy (HAADF-STEM) images of AgCrS₂, AgBiSe₂, and CuInS₂, scale bar: 0.5 nm. The insets display the side view of the atomic structure models and the fast Fourier transform (FFT) patterns. The blue and red polygons represent the octahedral and tetrahedron sites. **c, f, i** Intensity profiles of the orange dash lines in (**b**), (**e**), and (**h**).

synthesis method. Compared to other ion conductors, the ionic conductivity of the AgCrS₂ nanosheets synthesized by us remains at a relatively high level (Fig. 4c)[16,40–46].

Cu⁺/Ag⁺ ions can not only undergo long-distance migration but also experience local displacements, giving rise to ferroelectricity, specifically ion displacement-induced ferroelectricity[47]. By applying a vertical electric field, we can drive ion displacements between the two enantiomeric tetrahedral sites within the MX₂ interlayer space, resulting in spontaneous polarization reversal and exhibiting ferroelectric properties (Fig. 4d). Taking CuScS₂ as an example. The SHG mapping shows the uniform non-centrosymmetric crystal structure and the single domain characteristics of the as-synthesized CuScS₂ flake (Supplementary Fig. 41). Then, we employed piezoresponse force microscopy (PFM) testing methods to investigate the material's room-

temperature ferroelectric properties. The as-grown CuScS₂ flakes exhibited distinct ferroelectric phase hysteresis loops and typical amplitude butterfly curves (Fig. 4e). Simultaneously, we performed domain read-write operations on the flakes. The PFM phase image shows two stable opposite polarization domain regions (the inset of Fig. 4e), validating the out-of-plane ferroelectric properties of CuScS₂ at room temperature. We also tried to reveal the in-plane (IP) polarization of the CuScS₂ flakes. However, there is no typical IP PFM phase and amplitude hysteresis loop (as Supplementary Fig. 42 shows), suggesting no IP polarization in the as-synthesized CuScS₂ flakes. Furthermore, to reveal the temperature stability of the ionic displacement-type ferroelectricity in CuScS₂, which is critical to the applications, we employed high-temperature SHG measurements to characterize its ferroelectric Curie temperature ($T_C$). As the

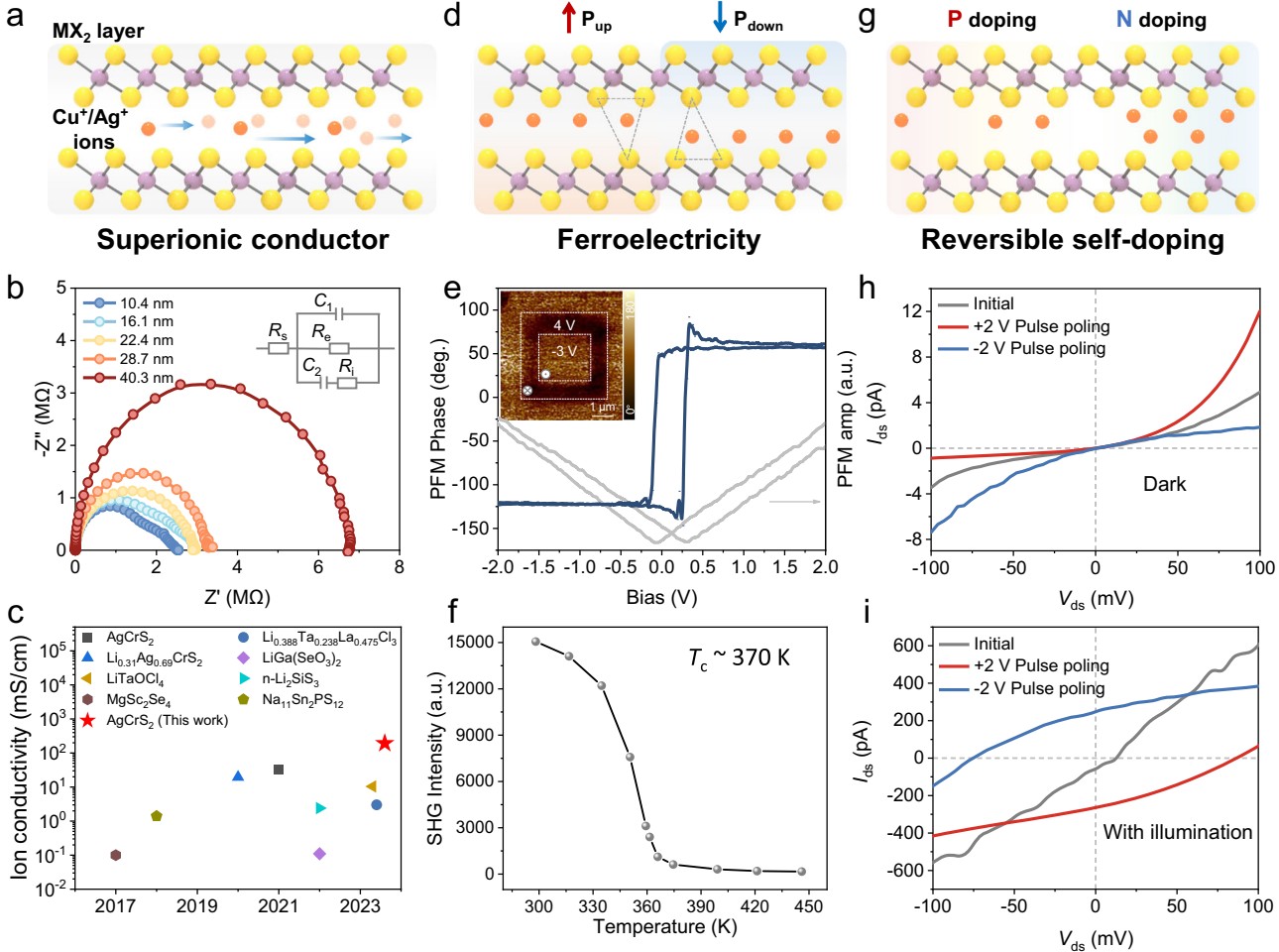

**Fig. 4 | The ionic and electronic properties of two representative as-synthesized 2D AMX$_2$ compounds. a, d, g** Schematic diagrams of the superionic conductor (**a**), ferroelectricity (**d**), and reversible self-doping (**g**) properties that arise from the ionic-electronic coupling effects. The blue arrows in (**a**) present the directed long-distance migration of the Cu$^+$/Ag$^+$ ions within the interlayer. The dashed triangles in (**d**) show the two enantiomeric tetrahedral sites of Cu$^+$/Ag$^+$ ions, corresponding to the up and down ferroelectric polarization states of the materials. **b** The impedance spectroscopic measurement for AgCrS$_2$ nanosheets with different thicknesses. The inset shows the equivalent circuit, where $R_s$, $R_e$, and $R_i$ correspond to the contact resistance, electronic resistance, and ionic resistance, respectively, and $C_1$ and $C_2$ are the constant phase elements. The detailed calculation process of ionic conductivity is shown in Supplementary Fig. 39. **c** The comparison of ionic conductivity with other reported superionic conductors[16,40–46]. **e** Piezoresponse force microscopy (PFM) phase and amplitude hysteresis loop of CuScS$_2$ nanosheet, the inset is the ferroelectric domains (white dashed square area) after forward and reverse DC bias polarization. The out-of-plane arrow symbols represent the P$_{up}$ and P$_{down}$ states of the nanosheet after polarization. **f** Temperature-dependent optical second harmonic generation (SHG) measurement of a CuScS$_2$ nanosheet, demonstrating a ferroelectric Curie temperature ($T_c$) of ∼ 370 K. **h, i** Memristor behavior (**h**) and switchable photovoltaic behavior under illumination ($\lambda$ = 532 nm, 256.6 mW/cm$^2$) (**i**) after positive/negative pulse polarization of the CuScS$_2$-based device.

temperature increases to above 370 K, the SHG signal of CuScS$_2$ nearly quenches (Fig. 4f and Supplementary Fig. 43a), indicating the disruption of the ordered occupancy of Cu$^+$ ions in the interlayer tetrahedral positions (Supplementary Fig. 43b). This implies that the $T_C$ of CuScS$_2$ is approximately 370 K, demonstrating significant potential for extensive applications in the field of ferroelectricity.

The movement of ions within the material often results in concentration gradients, leading to different doping effects[48]. By employing electrical pulses to control the migration of ions within the material, reversible self-doping characteristics can be achieved (Fig. 4g), consequently leading to interesting electrical and optoelectronic properties, which is exemplified by 2D CuScS$_2$. First-principles calculations predict that the migration barrier for a Cu$^+$ ion to move through the tetrahedral-octahedral-tetrahedral path within the CuScS$_2$ is 0.24 eV (Supplementary Fig. 44). Such a low migration energy barrier is comparable to that for intercalated Li$^+$ ions in transition metal dichalcogenides (TMDs)[49], indicating that a relatively small external electric field is sufficient to drive the migration of Cu$^+$ ions. Applying

electric pulses for polarization is an effective approach for achieving controlled migration of interlayer ions as we can modulate the pulse width and amplitude precisely[50]. Firstly, we investigated the ionic-electronic coupled properties of CuScS$_2$-based devices under dark conditions using triangular electric pulses (Supplementary Fig. 45a). As shown in Fig. 4h, the initial state I-V curve of the device is symmetric in the positive and negative voltage ranges. After poling the device with a forward bias pulse, the I-V curve exhibits characteristic diode-like behavior, indicating the generation of a potential barrier within the device under the influence of the electric pulse. Furthermore, upon applying a reverse bias pulse, the rectification direction of the device is reversed, indicating that the direction of internal potential barriers within the material can be modulated by the electric pulse. This reconfigurable potential barrier allows for continuous modulation of the device resistance through electric pulses, enabling memristive functionality (Supplementary Fig. 45b–d). Under illumination, compared to the initial state, the device exhibits a noticeable photovoltaic response after poling by 2 V/0.5 s electric pulse. The device

demonstrates a photovoltaic short-circuit current ($I_{sc}$) of ~270 pA when the source-drain bias voltage ($V_{ds}$) is zero and shows a distinct open-circuit voltage ($V_{oc}$) of ~85 mV. Similar to the dark situation, the photovoltaic response direction also reverses after the device undergoes reverse poling electric pulses (Fig. 4i). This reconfigurable photovoltaic response exhibits great stability during $I$-$t$ testing (Supplementary Fig. 46). To reveal the mechanism of this reconfigurable photovoltaic response, we firstly analyzed the possible bulk photovoltaic effect (BPVE) in the ferroelectric CuScS$_2$. Based on the single-domain characteristic of as-synthesized CuScS$_2$ (Supplementary Fig. 41), the BPVE would induce obvious $I_{sc}$ at the initial-state device. However, the $I_{sc}$ of the initial-state device is ignorable (Fig. 4i), demonstrating that the BPVE is negligible here. Moreover, the undetectable IP ferroelectricity of the as-synthesized CuScS$_2$ also proves this point. Then, we consider that the long-distance migration of the Cu$^+$ ions could induce reversible self-doping, which also can give rise to the reconfigurable photovoltaic response. Thus, we conducted cross-sectional EDS scanning tests on the polarized devices and observed that the distribution of Sc and S elements in the material remains uniformly distributed. However, the content of Cu elements beneath the electrode is significantly higher than in the channel region (Supplementary Fig. 47). This confirms the migration of Cu$^+$ ions after poling by the electric pulse. Moreover, the first-principles calculations indicate that the local absence or accumulation of Cu$^+$ ions resulting from ion migration will introduce significant p-type or n-type doping in the material (Supplementary Fig. 48), which is similar to the doping effect of point defects[5]. This coupling effect between ion migration and charge doping induces the emergence of the modulated potential barrier within 2D CuScS$_2$ and finally gives rise to the memristive behavior and reconfigurable photovoltaic response, which is crucial to the logic and neuromorphic devices[5,51].

In summary, we have demonstrated a separated-precursor-supply CVD method to control the reactions and vaporization of precursors. 20 kinds of 2D AMX$_2$ compounds have been synthesized successfully showing the practicality of our approach. Detailed structural analysis and comprehensive characterization have revealed the high crystalline quality of the prepared AMX$_2$ materials. Notably, the as-grown 2D AMX$_2$ flakes show intriguing ionic and electronic properties. A high ionic conductivity of 192.8 mS/cm can be observed in a 10.4 nm AgCrS$_2$ flake at room temperature. The synthesized 2D CuScS$_2$ flakes show ion displacement-induced ferroelectricity at room temperature. Meanwhile, the reconfigurable photovoltaic response based on the coupling of ions migration and charge doping also can be observed in CuScS$_2$. The achievement of generally synthesizing 2D AMX$_2$ compounds offers new insights into the vapor-phase synthesis of multielement 2D materials and provides an excellent material choice for exploration in electronics, optoelectronics, and neuromorphics.

## Methods
### CVD growth
2D AMX$_2$ compounds were synthesized by separated-precursor-supply CVD. Ultrahigh purity Ar (purity 99.999%) and H$_2$ (purity 99.999%) were used as the carrier gases. All the raw precursors were bought from Alfa Aesar with a purity higher than 99%. Freshly exfoliated mica was chosen as the growth substrate. A single temperature zone tube furnace (diameter, one inch) was used as the reaction instrument with a heating zone length of ~30 cm (Supplementary Fig. 5). A porcelain boat containing chalcogen precursor was placed upstream and heated to 150°C for sulfur (300°C for selenium). Two metal precursors were placed at the bottom of a one-side sealed small quartz tube with a length of 7 cm and a diameter of 10 mm. The growth substrate was placed in the small quartz tube too, 0.5-4 cm away from the metal precursors. Then, the small quartz tube was put downstream of the tube furnace and followed by the heating process. The heating rate is

50°C/min or 30°C/min for different AMX$_2$, and the reactions were carried out at 1 atm pressure. The schematic image of the CVD setup can be seen in Fig. 1c and Supplementary Fig 4a. The detailed growth parameters and descriptions are shown in Supplementary Table 1 and Supplementary Methods. The detailed characterizations of the morphology, phase, and atomic structures of as-synthesized ultrathin 2D AMX$_2$ are shown in Supplementary Figs. 11, 12, 14–38.

### CFD simulations
To reveal the mass flow during the experiment, we did the numerical finite element simulation. During the modeling, we followed the real experiment setup of the tube furnace, the details can be seen in Supplementary Fig. 5. The transient model is used to simulate the real CVD process. The gravity and convection heat transfer were considered. Argon and air are considered ideal gases. The shear stress transfer (SST) model is used here. It is a low Reynolds number and comprehensive turbulence model, which can give more accurate near-wall results[52]. Firstly, we conduct a steady analysis of the system. Similar to the real condition of the heating process of the furnace, we assume the system is stable when the temperature of the monitor point reaches the goal point and does not change again. Based on this result, the transient analysis was conducted to reveal the mass transportation process of two precursors.

### TG-DSC
TG-DSC testing was conducted using a DIAMOND TG/DTA thermal analyzer. Approximately 5 mg of the sample was added to an alumina crucible and heated from 20 °C to 900 °C at a rate of 50 K per minute in a pure argon atmosphere.

### Characterizations
The morphologies of the as-grown AMX$_2$ nanosheets were characterized by an optical microscope (BX51, OLYMPUS) and atomic force microscope (Dimension icon, Bruker). Raman, photoluminescence, and SHG spectra were obtained by a confocal Raman system (Alpha 300 R, WITec) equipped with a 532 nm CW laser and a high-temperature test chamber (TS1000EV, Linkam). Femtosecond laser (Verdi, Coherent) was applied as the excitation source of SHG measurement. For the cross-sectional HAADF-STEM and EDS measurements, the samples were prepared by focus ion beam (Helios NanoLab G3 UC, FEI). The atomic resolution HAADF-STEM images were obtained by a double CS-corrected transmission electron microscopy (Titan Themis G2 60-300, FEI). For the TEM measurements, the samples were prepared with a poly (methyl methacrylate) (PMMA) assisted transfer method. TEM, SAED, and EDS were performed on an FEI Tecnai G2 F30 instrument.

### Impedance spectra measurements
AgCrS$_2$ nanosheets were transferred to the silica substrate. Then, the Au/AgCrS$_2$/Au devices were fabricated by electron-beam lithography (EBL, FEI Quanta 650 SEM & Raith Elphy Plus) and thermal evaporation coating (Angstrom Engineering, Nexdep). The ionic conductivity of AgCrS$_2$ nanosheets is obtained by fitting the electrochemical impedance spectra (Autolab PGSTAT 302 N) of the Au/AgCrS$_2$/Ag devices at room temperature. The testing frequency range is 1 Hz to 1 MHz. To avoid electromagnetic interference the whole process was operated in a Faraday cage.

### PFM measurements
The PFM measurements were conducted on the AFM platform (Dimension icon, Bruker). CuScS$_2$ nanosheets were transferred to the silica substrate that was covered with gold. A DC voltage was applied to the conductive tip coated with Pt/Ir to reverse the ferroelectric domain of the sample.

## Device fabricating and electrical measurements

All the devices were transferred on the silicon substrate with a 300-nm-thick oxide film using a PMMA-assisted method. The Bi/Au was chosen to be the contact electrode. The electrical measurements were carried out with a semiconductor characterization system (4200SCS, Keithley) and a cryogenic probe station (CRX-6.5, Lake Shore).

## Theory calculations

The present first principle DFT calculations are performed by Vienna Ab initio Simulation Package (VASP)[53] with the projector augmented wave (PAW) method[54]. The exchange-functional is treated using the generalized gradient approximation (GGA) of Perdew-Burke-Ernzerhof (PBE)[55] functional. The energy cutoff for the plane wave basis expansion was set to 450 eV and the force on each atom less than 0.02 eV/Å was set for the convergence criterion of geometry relaxation. The Brillouin zone integration is performed using $5 \times 5 \times 5$ and $9 \times 9 \times 9$ k-point sampling for structure optimization and electronic structure calculation, respectively. The self-consistent calculations apply a convergence energy threshold of $10^{-5}$ eV. Transition state searching was performed using the climbing-image nudged elastic band (CI-NEB) method[56].

## Data availability

The data that support the plots within this paper and other findings of this study are available from the corresponding authors upon request.

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

## Acknowledgements

This work was supported by the National Natural Science Foundation of China (52322210, 22375069, 52172144, and U21A2069), the National Key Research and Development Program of China (2021YFA1200500), and the Innovation Project of Optics Valley Laboratory (OVL2023PY007). Here, the authors also thank the technical support from the Analytical and Testing Center at Huazhong University of Science and Technology.

## Author contributions

X.Z., T.Z., and X.X. conceived and designed the experiments. X.X. and Y.C. synthesized the materials. X.X., P.L., Y.C., and D.L. performed the AFM, Raman, and SHG characterizations of the samples. X.X. and H.L. performed the HRTEM and HAADF-STEM characterizations and worked on the analysis of the results. X.X. performed the electrochemical impedance measurements. X.X., P.L., Y.C., H.W., and X.S. performed device fabrication and measurement. X.X. and Z.L. performed the PFM measurements. X.X. and Y.C. wrote the paper with inputs from T.Z., X.Z., and J.W. All authors participated in discussions and approved the manuscript.

## Competing interests

The authors declare no competing interests.
