## [Peer Review File · Nature Communications]

General synthesis of ionic-electronic coupled two-dimensional materialsREVIEWER COMMENTS

Reviewer #1 (Remarks to the Author):

The authors present a novel chemical vapor deposition (CVD) method employing a separated-precursor-supply strategy, which effectively mitigates side reactions between metal precursors and excess chalcogen precursors, ultimately enabling the universal synthesis of 20 types of AMX₂ compounds. These synthesized AMX₂ compounds exhibit diverse crystal structures and demonstrate intriguing properties, including superionic conductivity, ferroelectricity, and reconfigurable self-doping characteristics. This work has yielded significant advancements in the preparation of multi-element two-dimensional materials and the exploration of materials exhibiting ionic-electronic coupled properties, which provide a totally new platform for transistors, photodetection, and neuromorphic computing. I think this work is very timely and can broadly attract intensive attention. Therefore, I strongly recommend this work to be accepted by Nature Communications after addressing the following questions:

1. Figure 2 illustrates the broad applicability of the synthesis strategy, wherein the synthesized 20 types of AMX₂ exhibit uniformly regular morphologies. It is observed that the majority of samples shows a triangular shape, while AgInS₂ presents a quadrilateral shape. Could the authors comment on the reasons underlying this morphology difference.
2. The authors utilized finite element simulation methods to intuitively reveal the chemical reaction environment during the CVD process, including flow fields and mass transfer processes. The reverse mass transfer of metal precursors in the small quartz tube, along with the lower concentration of chalcogen precursors near the metal source, ensure the stable supply of metal precursors, thereby facilitating the preparation of 20 types of 2D AMX₂ compounds. A comprehensive discussion of the reaction mechanism was provided. I am interested in whether the authors simulate the growth condition at different growth temperatures?
3. The synthesized 20 types of AMX₂ compounds contain 9 different metal elements, including two monovalent metal ions, Cu⁺ and Ag⁺, as well as trivalent metals like Sc³⁺, Cr³⁺, Fe³⁺, and Ga³⁺, etc. Within AMX₂ compounds, these metals can form bonds with chalcogen elements either in a tetrahedral coordination or in an octahedral coordination. This chemical bonding difference endows AMX₂ compounds with diverse structures and intriguing properties. Could the authors discuss the reasons behind this variation in chemical bonding?
4. The AMX₂ compounds showcased exhibit a rich variety of structures and manifest intriguing properties, including superionic conductivity, ferroelectricity, reconfigurable self-doping, etc. Two-dimensional ferroelectric materials are of significant importance in fields such as high-density storage and reconfigurable logic devices. I am curious whether, apart from CuScS₂, other AMX₂ compounds also exhibit ferroelectric properties?

Reviewer #2 (Remarks to the Author):

The authors successfully synthesized the 2D AMX₂ materials through the CVD method, by using a separated-precursor-supply route. Though similar 2D single crystals have been grown by CVT method, I am sure that it is hard to deposit by common CVD method, because of the multiple element components. I believe this general route itself is an important accomplishment, worthy of publication on its own; and the functional properties such as ionic conductivity and ferroelectricity are novel. I will suggest its publication, contingent upon some further clarification or explanation.

-the authors have put lots of synthesis data into the supplementary materials, which I will suggest to place them into the main text, to support the novelty the work. For example, Figures S10b & 10c shows that the proposed method can deposit the homogenous single phase CuCrSe₂ in a large area, which is very important when compared with traditional method shown in Figures S3b & 3c. In addition, the authors are suggested to show that they can have single phase in large area for other AMX₂ materials.

-the ferroelectricity of CuScS₂ is supported by the PFM domain switching and SHG excitation (Figure S40). The later can tell us the Curie temperature of the ferroelectric ordering, which is the key data and is suggested to be placed into the main text.

-the authors also show the switchable photovoltaic effect of CuScS₂ flakes controlled by opposite voltage poling, and they attributed it to the ion migration of Cu⁺ towards the electrode interface. Because CuScS₂ is a ferroelectric, and seems it has in-plane polarization as well from Figure S40. Ferroelectric polarization can induce intrinsic photovoltaic effect as well. One question raise immediately that does this in-plane polarization contribute to the switchable photovoltaic effect?

Reviewer #3 (Remarks to the Author):

Multi-element two-dimensional materials offer an additional degree of freedom by introducing an extra element, allowing for the modulation of material structure and properties, which is crucial for developing novel two-dimensional materials with superior performance. However, the synthesis of multi-element two-dimensional materials is currently limited by uncontrollable chemical reactions and precursor supply. This work proposed a separated-precursors-supply chemical vapor deposition strategy to ensure multiple metal precursors' stable supply and suppress the side reactions efficiently. Then, the universal synthesis of 20 types of ternary AMX₂ compounds was ultimately achieved. Furthermore, these materials exhibit intriguing ionic-electronic properties. This work not only offers an effective strategy for the synthesis of multi-element two-dimensional materials but also introduces a novel materials platform. Thus, I think this work is sufficiently innovative to be published in Nature Communications and suggest acceptance of this manuscript. Only some minor revisions are needed,

detailed comments see below:

1. AMX₂ materials exhibit rich and intriguing ionic-electronic coupled properties, including superionic conductivity, ferroelectricity, and reversible self-doping. These characteristics endow the materials with significant potential applications in fields such as energy storage and information technology. Could the authors comment on the challenges and opportunities to the synthesis of the AMX₂ for large-scale application?

2. As well known, because of the nuclear and growth processes of the products occurring on the substrate. Thus, substrate is a key factor in the growth process of target products. There are also many other kinds of substrates as SiO₂/Si, Al₂O₃, etc. for two-dimensional materials. Herein, the authors employed mica as the collected substrate. Thus, the authors are suggested to explain why choosing mica substrates.

3. The optical second harmonic generation (SHG) signal is related to the non-centrosymmetric nature of ferroelectric materials. In this work, the as-synthesized CuScS₂ flakes exhibit stable room-temperature ferroelectricity with a high ferroelectric Curie temperature of 370 K and has been verified by the temperature-dependent SHG measurements. Thus, the authors are suggested to provide the corresponding complete SHG spectra at different temperatures so that the readers can understand the novel ion-induced ferroelectricity better.

4. There are some minor formatting issues in the manuscript, such as the overlap between the character "1" and the x-axis in Figure 1g, in some figure captions, the initial letters are not capitalized, and missing scale bars in some AFM images in Supplementary Figure 11.

Response to Reviewer 1:

Reviewer #1: The authors present a novel chemical vapor deposition (CVD) method employing a separated-precursor-supply strategy, which effectively mitigates side reactions between metal precursors and excess chalcogen precursors, ultimately enabling the universal synthesis of 20 types of AMX_2 compounds. These synthesized AMX_2 compounds exhibit diverse crystal structures and demonstrate intriguing properties, including superionic conductivity, ferroelectricity, and reconfigurable self-doping characteristics. This work has yielded significant advancements in the preparation of multi-element two-dimensional materials and the exploration of materials exhibiting ionic-electronic coupled properties, which provide a totally new platform for transistors, photodetection, and neuromorphic computing. I think this work is very timely and can broadly attract intensive attention. Therefore, I strongly recommend this work to be accepted by Nature Communications after addressing the following questions:

Q1: Figure 2 illustrates the broad applicability of the synthesis strategy, wherein the synthesized 20 types of AMX_2 exhibit uniformly regular morphologies. It is observed that the majority of samples shows a triangular shape, while $AgInS_2$ presents a quadrilateral shape. Could the authors comment on the reasons underlying this morphology difference.

Answer: Thanks for the kind suggestion. Generally, the crystal face can be attributed to two aspects: thermodynamic and kinetic driving forces. When the precursors' feeding is stable and uniform during the CVD process, the morphologies of the as-grown crystals are determined by the Gibbs-Curie-Wulff law, which is the principle of surface energy minimization for predicting the equilibrium crystal shape. When a crystal face exhibits higher surface energy, it tends to grow faster, leading to a reduction in its surface area and ultimately resulting in the dominance of faces with lower surface energy (*Chem. Soc. Rev.* 2022, 51, 7327; *Nat. Commun.* 2019, 10, 756). For $AgInS_2$, we have revealed the orthorhombic phase by TEM and SAED, and due to the honeycomb-like structure of its (100) face, the crystal could show a quadrilateral shape (edge surfaces $\{(0-11), (011), (01-1), (0-1-1)\}$), and also can show a triangle shape (edge surfaces $\{(020), (01-2), (012)\}$).

Q2: The authors utilized finite element simulation methods to intuitively reveal the chemical reaction environment during the CVD process, including flow fields and mass transfer processes. The reverse mass transfer of metal precursors in the small quartz tube, along with the lower concentration of chalcogen precursors near the metal source, ensure the stable supply of metal precursors, thereby facilitating the preparation of 20 types of 2D AMX_2 compounds. A comprehensive discussion of the reaction mechanism was provided. I am interested in whether the authors simulate the growth condition at different growth temperatures?

Answer: Thanks for the kind suggestion. Temperature is a critical factor for CVD. The nucleation and growth of crystals are closely related to the temperature. Different AMX_2 have various proper growth temperatures. To provide more comprehensive support for our synthesis mechanism, we simulated the flow fields and mass transfer conditions around the small quartz tube at 600°C, 700°C, and 800°C. As Figure R1-1 shows, the flow field has no obvious change at 600°C-800°C. The reverse gas flow inside the small quartz tube always exists in this temperature range, which means that this method allows us to realize a stable supply of metal precursors when synthesizing the 20 AMX_2 .

Figure R1-1. The velocity distribution around the small quartz tube at different temperatures. a-c, Distribution of Velocity u in the small quartz tube at 600°C, 700°C, and 800°C respectively. b, The variation curve of Velocity u along the dashed line in (a)-(c).

Q3: The synthesized 20 types of AMX_2 compounds contain 9 different metal elements, including two monovalent metal ions, Cu^+ and Ag^+ , as well as trivalent metals like Sc^{3+} , Cr^{3+} , Fe^{3+} , and Ga^{3+} , etc. Within AMX_2 compounds, these metals can form bonds with chalcogen elements either in a tetrahedral coordination or in an octahedral coordination. This chemical bonding difference endows AMX_2 compounds with diverse structures and intriguing properties. Could the authors discuss the reasons behind this variation in chemical bonding?

Answer: Thanks for the kind suggestion. In the AMX₂ compounds, Sc³⁺, Cr³⁺, Bi³⁺, Sb³⁺ exhibit octahedral coordination, Fe³⁺, Ga³⁺, In³⁺ show tetrahedral coordination, and Ag⁺, Cu⁺ can either occupy the octahedral or tetrahedral sites. Due to the steric effects, the oxidation state and size of the metals are important to the coordination numbers. For more transition metals tend to form up to 6 bonds. According to crystal field theory, the energy level of the *e* in the tetrahedral coordination is higher than the *t_{2g}* band in the octahedral coordination (Figure R1-2). When the number of outmost electrons is less than 7, the filling of *d_{xy}*, *d_{xz}*, *d_{yz}* orbitals of octahedral is more favorable. Actually, most transition metals can be induced to form octahedral coordination and examples exist for every *d* electron configuration ranging from group-3 (*d*⁰) to group-12 (*d*¹⁰). Based on the hybrid orbital theory, *d*¹⁰ metals like Cu⁺, Ga³⁺, etc. have fully filled *d* orbital, left one *ns* orbital and three *np* orbitals to form four *sp*³ hybrid orbitals, resulting in the preference of tetrahedral coordination. Some heavier group-15 elements like Sb³⁺ and Bi³⁺ can form *d*²*sp*³ hybrid orbitals, which have octahedral coordination. In particular, the Cu⁺ and Ag⁺ exhibit the obvious Jahn-Teller effect, which can lower energy difference between different bonding configurations making its various occupations in AMX₂ compounds.

Figure R1-2. Schematic images of octahedral and tetrahedral coordination and energy levels of *d*-orbital electrons based on the crystal field theory.

Q4: The AMX₂ compounds showcased exhibit a rich variety of structures and manifest intriguing properties, including superionic conductivity, ferroelectricity, reconfigurable self-doping, etc. Two-dimensional ferroelectric materials are of significant importance in fields such as high-density storage and reconfigurable logic devices. I am curious whether, apart from CuScS₂, other AMX₂ compounds also exhibit ferroelectric properties?

Answer: Thanks for the kind suggestion. In the AMX₂, the origin of ferroelectricity is mainly from the non-centrosymmetric lattice occupation of Cu⁺ and Ag⁺. The electric

field can polarize the interlayered Cu^+ from tetrahedral site α to β in CuScS_2 , thus switching the ferroelectric polarization, thus showing ferroelectric properties. So, the AMX_2 which possesses a similar layered structure characteristic of CuScS_2 , such as CuCrS_2 , CuCrSe_2 , AgCrS_2 , etc., may exhibit ferroelectricity. In addition, Wu. et al. theoretically predicted that the CuCrS_2 and CuCrSe_2 could show ferroelectricity when the thickness is down to nanometers (*Nat. Sci. Rev.* 2020, 7, 373). Additionally, we show PFM characterizations of some as-grown AMX_2 in Figure R1-3.

Figure R1-3. Ferroelectric polarization switching by PFM on CuCrS_2 and AgCrS_2 . The PFM phase image of a 15 nm CuCrS_2 (a) and a 10.8 nm AgCrS_2 nanosheet (c), scale bar: 1 μm . The corresponding PFM phase hysteresis and butterfly loops of a 15 nm CuCrS_2 (b) and a 10.8 nm AgCrS_2 nanosheet (d).

Response to Reviewer 2:

Reviewer #2: The authors successfully synthesized the 2D AMX_2 materials through the CVD method, by using a separated-precursor-supply route. Though similar 2D single crystals have been grown by CVT method, I am sure that it is hard to deposit by common CVD method, because of the multiple element components. I believe this general route itself is an important accomplishment, worthy of publication on its own; and the functional properties such as ionic conductivity and ferroelectricity are novel. I will suggest its publication, contingent upon some further clarification or explanation.

Q1: The authors have put lots of synthesis data into the supplementary materials, which I will suggest to place them into the main text, to support the novelty the work. For example, Figures S10b & 10c shows that the proposed method can deposit the homogenous single phase $CuCrSe_2$ in a large area, which is very important when compared with traditional method shown in Figures S3b & 3c. In addition, the authors are suggested to show that they can have single phase in large area for other AMX_2 materials.

Answer: We sincerely appreciate the valuable comments. We have reorganized our Figure 1. Some important synthesis data have been added in the latest version (Figure R2-1). And, to better show the uniform synthesis of AMX_2 , we add the larger area optical images of all 20 AMX_2 in the Supporting information (Figure R2-2).

Figure R2-1. Synthesis mechanism of CVD growth AMX_2 . a, The formation energy

of CuCrSe_2 phase diagram⁵⁷. **b**, The kinetic growth process is influenced by the supply of metal precursors. **c**, Schematic image of the CVD setup. **d**, The large area optical image of the as-synthesized CuCrSe_2 nanosheets. **e**, The Raman spectra of the flakes in the (d). **f**, The CFD simulated distribution of X (S/Se) and metal precursors concentration. **g**, The variation curve of precursor concentration along the line in Figure (e). **h**, TGA of CuI and Cu_2Se powders.

Figure R2-2. Typical optical images of the 20 kinds of 2D AMX_2 on the substrates, scale bar: 10 μm .

Q2: The ferroelectricity of CuScS_2 is supported by the PFM domain switching and SHG excitation (Figure S40). The later can tell us the Curie temperature of the ferroelectric ordering, which is the key data and is suggested to be placed into the main text.

Answer: Thanks for the kind suggestion. The ferroelectric Curie temperature reflects the stability of the ferroelectric ordering. To help readers get a better understanding, we have added the temperature-dependent SHG data in the revised manuscript(Figure R2-3).

Figure R2-3. The ionic and electronic properties of two representative as-synthesized 2D AMX_2 compounds. **a, d, g**, Schematic diagrams of the superionic conductor (**a**), ferroelectricity (**d**), and reversible self-doping (**g**) properties that arise from the ionic-electronic coupling effects. **b**, The impedance spectroscopic measurement for $AgCrS_2$ nanosheets with different thickness. **c**, The comparison of ionic conductivity with other reported superionic conductors. **e**, The PFM phase and amplitude hysteresis loop of $CuScS_2$ nanosheet, the inset is the ferroelectric domains of $CuScS_2$ nanosheet after forward and reverse DC bias polarization. **f**, The temperature-dependent SHG measurement of a $CuScS_2$ nanosheet. **h, i**, Memristor behavior (**h**) and switchable photovoltaic behavior under illumination ($\lambda=532$ nm, 256.6 mW/cm²) (**i**) after positive/negative pulse polarization of the $CuScS_2$ -based device.

Q3: The authors also show the switchable photovoltaic effect of $CuScS_2$ flakes controlled by opposite voltage poling, and they attributed it to the ion migration of Cu^+ towards the electrode interface. Because $CuScS_2$ is a ferroelectric, and seems it has in-plane polarization as well from Figure S40. Ferroelectric polarization can induce intrinsic photovoltaic effect as well. One question raise immediately that does this in-plane polarization contribute to the switchable photovoltaic effect?

Answer: Thanks for the kind suggestion. Based on the atomic structure of the $CuScS_2$, the Cu^+ will take place a small lateral displacement along with the vertical displacement when switching the ferroelectric polarization of the $CuScS_2$, which may induce the in-plane (IP) polarization. So, we have tried our best to investigate the IP polarization of the as-grown $CuScS_2$ nanosheets. However, maybe the IP polarization is too weak to be detected by the PFM, as Figure R2-4 shows, there is no typical IP

PFM phase and amplitude hysteresis loop. When discussing the mechanism of the switchable photovoltaic behavior of CuScS_2 devices, it is important to consider that there is no obvious short current when the device is at its initial state, which means the bulk photovoltaic effect is negligible, representing the small contributions of the IP polarization of the CuScS_2 . On the other hand, due to the low migration barrier (0.24 eV/atom), the Cu^+ ions prefer the long-distance migration under in-plane electric polarization, which has been proved by the cross-sectional EDS mapping of CuScS_2 device after polarization (Figure S45).

Figure R2-4. The OOP (a) and IP (b) PFM phase and amplitude hysteresis loop of a 38 nm CuScS_2 nanosheet.

Response to Reviewer 3:

Reviewer #3: Multi-element two-dimensional materials offer an additional degree of freedom by introducing an extra element, allowing for the modulation of material structure and properties, which is crucial for developing novel two-dimensional materials with superior performance. However, the synthesis of multi-element two-dimensional materials is currently limited by uncontrollable chemical reactions and precursor supply. This work proposed a separated-precursors-supply chemical vapor deposition strategy to ensure multiple metal precursors' stable supply and suppress the side reactions efficiently. Then, the universal synthesis of 20 types of ternary AMX_2 compounds was ultimately achieved. Furthermore, these materials exhibit intriguing ionic-electronic properties. This work not only offers an effective strategy for the synthesis of multi-element two-dimensional materials but also introduces a novel materials platform. Thus, I think this work is sufficiently innovative to be published in Nature Communications and suggest acceptance of this manuscript. Only some minor revisions are needed, detailed comments see below:

Q1: AMX_2 materials exhibit rich and intriguing ionic-electronic coupled properties, including superionic conductivity, ferroelectricity, and reversible self-doping. These characteristics endow the materials with significant potential applications in fields such as energy storage and information technology. Could the authors comment on the challenges and opportunities to the synthesis of the AMX_2 for large-scale application?

Answer: Thanks for your kind suggestion. Developing large-scale array devices is crucial for practical applications in the future. The development of CVD synthesis methods for large-area 2D AMX_2 plays a key role. Due to the crystal structures of some AMX_2 are composed of $[CrS_2]$, $[ScS_2]$, $[ScSe_2]$, and $[BiSe_2]$, etc., we can draw inspiration from the recent advances in the preparation of wafer-scale transition metal dichalcogenides, for example the substrate-engineering strategies (*Chem. Soc. Rev.* 2023, 52, 1650). However, ensuring precise control over stoichiometry in large-scale synthesis still be challenging due to the issues such as non-uniform mixing and variations in reaction conditions. Combining the separated-precursor-supply idea of this work and substrate-engineering strategies, we may realize the wafer-scale synthesis of the AMX_2 materials in the future. On the other hand, we believe these novel 2D AMX_2 will attract intensive attention in photoelectronic artificial neural networks and integrated sensing-computing devices based on their controllable ion migration properties.

Q2: As well known, because of the nuclear and growth processes of the products occurring on the substrate. Thus, substrate is a key factor in the growth process of target products. There are also many other kinds of substrates as SiO_2/Si , Al_2O_3 , etc. for two-dimensional materials. Herein, the authors employed mica as the collected substrate. Thus, the authors are suggested to explain why choosing mica substrates.

Answer: Thanks for the kind suggestion. Although SiO₂/Si and Al₂O₃ have been used as the collected substrate for several two-dimensional structures, they are not the ideal substrates for all the two-dimensional structures, especially for the non-van der Waals structure materials. The reason for choosing mica substrates is that layered mica substrate is considered to be a perfect van der Waals epitaxy substrate for growing 2D materials due to the atomically smooth surface. It not only avoids the strict requirement of lattice matching between the epitaxial two-dimensional material and substrates but also facilitates the fast migration of the adatoms during the growth (*Adv. Funct. Mater.* 2018, 28, 1800181; *Adv. Mater.* 2017, 29, 1703122).

Q3: The optical second harmonic generation (SHG) signal is related to the non-centrosymmetric nature of ferroelectric materials. In this work, the as-synthesized CuScS₂ flakes exhibit stable room-temperature ferroelectricity with a high ferroelectric Curie temperature of 370 K and has been verified by the temperature-dependent SHG measurements. Thus, the authors are suggested to provide the corresponding complete SHG spectra at different temperatures so that the readers can understand the novel ion-induced ferroelectricity better.

Answer: Thanks for the kind suggestion. We have provided every single SHG spectrum at different temperatures in the revised supporting information.

Figure R3-1. Temperature-dependent SHG measurement of an as-grown CuScS₂ nanosheet. a, The spectra of SHG signal at different temperatures. b, Schematic diagram of the arrangement of Cu⁺ ion in CuScS₂ structure when the temperature is below and beyond the T_c .

Q4: There are some minor formatting issues in the manuscript, such as the overlap between the character "1" and the x-axis in Figure 1g, in some figure captions, the initial letters are not capitalized, and missing scale bars in some AFM images in Supplementary Figure 11.

Answer: We are sorry for the careless mistakes. We have checked the whole manuscript again and corrected the problems. The changes have been highlighted in red.

List of changes (highlighted in red)

1. Page 6 line 11-12: we added the statement and citation of Fig. 1d and 1e;
2. Page 7 line 6: we changed the citation of Fig. 1h;
3. Page 7 line 17: we added the citation of Fig. 1d and 1d;
4. Page 7 line 18-19: we added the statement and citation of Supplementary Fig. 11;
5. Page 12 line 3: we added the citation of the inset Figure of Fig. 4e;
6. Page 12 line 8: we added the citation of Fig.4f;
7. Page 26 line 2: we revised the Fig. 1;
8. Page 29 line 2: we revised the Fig. 4;
9. Supplementary Information, Page 2 line 8 and line 21: we revised the figures' title;
10. Supplementary Information, Page 9 line 2: we revised the Fig. S6;
11. Supplementary Information, Page 13 line 2: we revised the Fig. S10;
12. Supplementary Information, Page 20 line 3: we added the Fig. S11;
13. Supplementary Information, Page 21 line 2: we revised the Fig. S12;
14. Supplementary Information, Page 52 line 2: we revised the Fig. S41.

REVIEWERS' COMMENTS

Reviewer #1 (Remarks to the Author):

The authors have addressed concerns adequately, I think the manuscript would be appropriate for publication.

Reviewer #2 (Remarks to the Author):

The authors have clarified several key issues in their response to the reviewers and have improved their manuscript accordingly. The revision is impressive. Regarding on the correlation between in-plane ferroelectric polarization and switchable photovoltaic effect, the authors response is appropriate, though they failed in revealing the in-plane polarization. The authors claimed that the bulk photovoltaic effect is minor because of the lack of short circuit current in the initial state, which, however, may also be due to the as-grown multi-domain state. Whatever, I will suggest to include this part into the main text to give a comprehensive picture of this compound before publication.

Response to Reviewers' Comments

Response to reviewer #1:

The authors have addressed concerns adequately, I think the manuscript would be appropriate for publication.

Response: We thank the reviewer for recommending our work for publication.

Response to reviewer #2:

The authors have clarified several key issues in their response to the reviewers and have improved their manuscript accordingly. The revision is impressive. Regarding on the correlation between in-plane ferroelectric polarization and switchable photovoltaic effect, the authors response is appropriate, though they failed in revealing the in-plane polarization. The authors claimed that the bulk photovoltaic effect is minor because of the lack of short circuit current in the initial state, which, however, may also be due to the as-grown multi-domain state. Whatever, I will suggest to include this part into the main text to give a comprehensive picture of this compound before publication.

Response: We thank the reviewer for recommending our work for publication. To address the concern of the as-grown multi-domain state, we provide the optical second harmonic generation (SHG) mapping data. Optical SHG is believed to be sensitive to the grain boundary, domain wall, strain, defects, etc. (*Adv. Funct. Mater.* **2022**, *32*, 2105259., *Phys. Rev. Lett.* **2018**, *120*, 227601.). Specifically, if the as-grown flake is multi-domain, due to the disappearance of the symmetry breaking at the domain wall, the SHG mapping image will show the domain wall where the signal is zero. However, as Figure R1 shows, the uniform SHG signal distribution of the as-grown CuScS₂ nanosheet demonstrates the single ferroelectric domain characteristic. This result also gives proof of the high quality of the as-grown materials. As the reviewer's kind suggestion, we have added a proper discussion about this part to help readers better understand the properties of the materials.

Fig. R1. The SHG mapping of an as-synthesized CuScS₂ nanosheet. SHG signal is sensitive to the local crystal structure, it can be used to reveal the crystals' crystallinity, defects, domain wall, etc. The uniform distribution of the SHG signal demonstrates the good crystallinity and the single-domain characteristics of the as-synthesized CuScS₂.